# Dissecting the Genetic Mechanisms of Hemicellulose Content in Rapeseed Stalk

**Yinhai Xu [1], Yuting Yang [2], Wenkai Yu [2], Liezhao Liu [3], Qiong Hu [2]**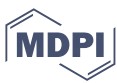**, Wenliang Wei [1],\* and Jia Liu [2],\***

1　College of Agriculture, Yangtze University, Jingzhou 434000, China
2　Oil Crops Research Institute, Chinese Academy of Agricultural Sciences, Wuhan 430000, China
3　College of Agronomy and Biotechnology, Southwest University, Chongqing 400715, China
\*　Correspondence: whwenliang@163.com (W.W.); liujia02@caas.cn (J.L.); Tel.: +86-0716-806-6314 (W.W.); +86-027-8671-1556 (J.L.)

**Abstract:** Polysaccharides such as hemicellulose in rapeseed can be used as an abundant resource to develop biomass energy. In the present study, the hemicellulose content in the middle stalk and taproot of a rapeseed core population of 139 accessions in Guizhou, Hubei and Anhui provinces was determined. Genotyping of the core population was carried out by a 60 K single nucleotide polymorphism chip, and a genome-wide association study (GWAS) was performed to reveal the associated sites of hemicellulose content in rapeseed. The results of the GWAS showed that 28 SNPs ($p \leq 0.001$) were significantly associated with hemicellulose content, and revealed that three sites—qHCs.C02 (contribution rate = 17.20%), qHCs.C05 (10.62%), and qHCs.C08 (8.80%)—are significantly associated with hemicellulose content in the stalk and three sites—qHCt.A09 (9.49%), qHCt.C05 (9.18%) and qHCt.C08 (13.10%)—are significantly associated with hemicellulose content in the taproot. Seven candidate genes associated with hemicellulose synthesis were identified in these major loci. Further RNA-seq analysis showed that two key differentially expressed genes (*BnaC05G0092200ZS* and *BnaC05G0112400ZS*) involved in hemicellulose synthesis were identified as having underlying QTL. This study excavated the key loci and candidate genes for regulating hemicellulose synthesis, providing a theoretical basis for developing rapeseed varieties with high hemicellulose content. At the same time, our results will be helpful in producing rapeseed cultivars with high lodging-resistance as well as highlighting the value of rapeseed as a resources for the bioenergy industry.

**Keywords:** rapeseed; hemicellulose; GWAS; RNA-seq; bioenergy

## 1. Introduction

Plant cell walls are composed of many polymers, including cellulose, hemicellulose, pectin, lignin and cell wall proteins [1]. Cell walls are the most basic structures of plant cells, and can maintain the mechanical strength of plant stems and control cell growth and the transmission of intercellular signal substances. They are also one of the most complex network structures in nature [2]. Hemicellulose, which is one of the three main components of the cell wall, is synthesized in the Golgi apparatus and deposited on the surface of cell wall through vesicles. Hemicellulose is a type of heterogeneous polymer composed of different types of monosaccharides, including xylan, xyloglucan and galactomannan. Among these monosaccharide components are glucose, xylose, arabinose and galactose, which are connected by small groups such as acetyl, phenolic acid, ferulic acid and coumaric acid in the main chain or side chains [3]. In different stages of plant-tissue and cell development, the hemicellulose skeleton is often replaced by different glycosyl residues in certain patterns, which determine the changes in physicochemical properties and structure of hemicellulose skeleton, leading to differences in its molecular chemical structure [4]. Five different hemicellulose models have been simulated [5] by means of molecular dynamics and their interaction characteristics and their bonding strength with

cellulose in the formation of nanocrystals analyzed. It is believed that, the interaction arrange $H_2O$, hemicellulose and the cellulose surface may be weakened by the hydrogen bond of the hydroxyl on the cellulose surface [5]. Hemicellulose is synthesized in the Golgi apparatus and deposited on the surface of cell wall through vesicles, and it can be dissolved in alkaline solution [1]. After encountering an acidic solution, it is more prone to hydrolyze than cellulose, so it is easier to process and degrade.

Hemicellulose exists widely in plants, and mainly plays the role of skeleton support. Hemicellulose accounts for about 15% to 20% of coniferous wood, and the main component is polygalactose mannose. It also accounts for about 15% to 35% of broad-leaved wood and grasses, and its main component is xylan [6]. In crops, the hemicellulose components in barley grains are mainly a mixture of xylo-oligosaccharide and arabinose [7]. The hemicellulose components in wheat stalks and rice stalks are mainly arabinoxylan, and the main chain is xylose, while the branches are composed of arabinose and glucuronic acid [8]. The hemicellulose in konjac tubers is mainly mannose poly-grape, which has very high medicinal and edible value. It was found that hemicellulose could improve fruit firmness and quality during the storage process of blueberries [9], and the introducing of specific genes related to inhibition of hemicellulose synthesis through transgenic technology could control fruit deterioration, reduce spoilage rate and prolong shelf life [10]. This has important commercial significance. In cassava stem, with the lignification of the tissue parts, the hemicellulose structure in the stem also changes differently, which makes the polysaccharides more abundant in the body [11]. In addition, the study also found that hemicellulose biosynthesis had an impact on grain production and the environment. Aluminum toxicity and other heavy metals in soil can rapidly inhibit crop root elongation, which can significantly reduce crop yield. The *XTH31* gene has been found to regulate the content of xyloglucan in cell-wall hemicellulose as well as the aluminum sensitivity of *Arabidopsis* [12,13]. In ramie, an increase in the hemicellulose content of cell wall can enhance its ability to endure cadmium stress [14]. As a result, the hemicellulose in the cell wall is able to adsorb a large amount of aluminum and cadmium in soil, thereby reducing the content of heavy metals in the soil, effectively reducing heavy-metal stress and improving crop yield. Therefore, reducing the content of heavy metals in soil by increasing the hemicellulose content in crop roots is helpful in improving crop yield and quality. The study of hemicellulose content in crops is of great significance in agriculture, the environment and industry.

Hemicellulose has a high value as a resource in the chemical and bioenergy industries. Due to its good biodegradability, biocompatibility and bioactivity, hemicellulose can form various downstream compounds through various chemical, physical and biodegradation means, which can be directly or indirectly used in many bio-based industries and play an indispensable role in different industries [15]. For example, hemicellulose acts as a plasticizer and drug-delivery agent in the pharmaceutical industry. In the food industry, it is often used to produce the artificial sweetener xylitol [16]. In the chemical industry and energy production, hemicellulose can be used to produce the important biofuel ethanol to replace fossil fuels [17], as well as furfural, which is the precursor of maleic anhydride, oxalic acid, furfuryl alcohol and tetrahydrofuran [18]. The diversified and comprehensive utilization of hemicellulose has extremely value in a range of industries including the bioenergy industry.

Studying the genetic basis of controlling hemicellulose content in crops is an important basic work for improving and utilizing abundant hemicellulose resources in plants. In order to quickly and accurately determine the content of hemicellulose with lower cost and less labour, a near-infrared spectral model was developed [19]. In wheat, five QTL loci controlling seed xylan content were located by IciMapping software, and it was found that seed hardness significantly determined seed xylan content [20]. Linkage mapping and association analysis of fiber-quality-related traits in soybean were carried out by using a recombinant inbred line (RIL) population and a natural population, and three QTL loci were found to be significantly correlated with hemicellulose content [21]. Through

mapping the correlation between forage digestibility and hemicellulose content in spring and autumn, it was found that hemicellulose concentration of ryegrass was significantly correlated with spring digestibility [22]. Ma et al. (2013) analyzed hemicellulose content in rapeseed seeds by near-infrared spectroscopy, and detected eight QTL loci, and also confirmed that hemicellulose was positively correlated with protein content [23]. Overall, hemicellulose content is affected by many factors, such as grain hardness, fiber quality, seasonal digestibility and the environment.

High hemicellulose content has been shown to significantly increase the lodging resistance of crops [24]. Therefore, the key genes that can increase the hemicellulose content in rapeseed have been excavated, which will greatly improve the utilization of biomass energy and the lodging resistance of rapeseed. Rapeseed is an important and valuable oil crop in China as it has multiple uses, such as in honey, feed, fertilizer and ornamental plants, as well as in soil improvement [25]. Rapeseed is mainly used to produce edible oil and crude meal rich in high-value protein which can be used as livestock feed. Rapeseed stalk is the surplus of rapeseed after harvest, and can be used as source of livestock feed and bioethanol production as its crude-fiber content can be as high as 40% [26,27]. In the context of the increasing demand for fossil energy and decreasing available land, the production of alternative fuels using the rich straw wastes of rapeseed and other crops will play an important role in realizing the national strategy of "carbon compliance" and "carbon neutrality" [28].

The genetic basis of the hemicellulose content in rapeseed stalk and taproot has seldom been reported. In the present study, the content of hemicellulose in the stalk and taproot of rapeseed core germplasm resources in three environments was determined by near-infrared spectrometry. It was found that there were significant genetic differences in hemicellulose content among various environments. A genome-wide association study was used to identify the sites on the genome that significantly affected hemicellulose content in rapeseed. Two key candidate genes *BnaC05G0092200ZS* and *BnaC05G0112400ZS* associated with hemicellulose synthesis were obtained by analyzing transcriptome data of the rapeseed stem. The results will provide a theoretical basis for follow-up research on developing rapeseed germplasm resources, improving rapeseed lodging resistance and even facilitating the bioenergy utilization of rapeseed stalks.

## 2. Materials and Methods

### 2.1. Plant Material

The experimental materials were composed of 139 rapeseed (*Brassica napus* L.) core germplasm resources [29] (Supplementary Table S4), which were provided by the Oil Crops Research Institute, Chinese Academy of Agricultural Sciences.

### 2.2. Experimental Design

In order to obtain the phenotypic data of the three environments, the materials were planted in three experimental stations representing the upper, middle and lower reaches of the main rapeseed producing areas in the Yangtze River Valley, namely Zunyi City, Guizhou Province (28° N, 107° E), Yangluo City, Hubei Province (30° N, 114° E) and Lu'an City, Anhui Province (32° N, 116° E). The temperature in the three environments is stable and the fluctuation trend is consistent (Supplementary Figure S1). A random block design was adopted, and two replicates were set. In every three rows of each block, the length of each row was 1.8 m, and the intervals between rows were 0.33 m, while 15 individuals were planted in each row with standard field-planting management.

### 2.3. Phenotypic Data Collectionion

At the mature stage of the rapeseed plants, samples were taken from the stalk at 60 cm above the ground and from the taproot at 5 cm below the ground. Samples were taken from individuals with the same status in the middle row of each block. After natural storage and air drying, and oven drying to constant weight, the samples were ground into



powder and the hemicellulose content of each samples was evaluated by an established model [23] based on the spectral data obtained with a near-infrared instrument (Foss2500) at Southwest University.

### 2.4. Statistical Analysis

Descriptive statistics of trait data were completed by the R language (3.3.3) [30] and the hemicellulose content distribution maps in different environments were drawn using ggplot2 and GGally's r language software package [31,32]. The variation range, variation value and variance analysis of hemicellulose content in stalks and taproots of various environments were also calculated. Significance for the effect of genotype (G), environment (E) and genotype by environment interaction (G × E) on phenotypic variance was estimated by ANOVA across the three environments ($p < 0.05$ indicates significant; $p < 0.01$ indicates highly significant).

### 2.5. Genotyping, Population Structure Linkage and Disequilibrium Analysis

The Brassica Illumina 60K chip was used to obtain the genotypic data of 139 core germplasm resources of rapeseed. After quality control (minor allele frequency > 0.05, missing data < 20% [33]), single nucleotide polymorphisms (SNPs) were obtained for subsequent GWAS analysis, and 21426 SNPs were obtained [29]. The principal component analysis software Tassel 5.0 was used to obtain two principal component data, and then the linear correlation distribution maps of hemicellulose content in 139 rapeseed samples in different environments and different tissue parts were drawn by the ggplot2 package of R3.3.3 [31]. PopLDdecay software was use to draw the LD attenuation diagram to show the degree of chromosome linkage disequilibrium [34].

### 2.6. Genome-Wide Association Study

A Multi-environment mixed linear model (MLM) in Tassel 5.0 software was used for association analysis based on the phenotypic and genotypic data of 139 rapeseed samples [35]. K matrix, which represented the genetic relationship between the individuals in the association population, was added to the model to eliminate false positives caused by genetic relationship in the population. In the present study, the *p* value indicated whether a SNP was associated with the corresponding traits, and the contribution rate indicated the phenotypic variation explained by this marker. The standard for determining significant loci based on GWAS results was $p < 0.001$. The R (3.3.3) qqman software package was use to draw the Manhattan map and the QQ map [36].

### 2.7. Functional Annotation and Tissue Expression Analysis of Candidate Genes

According to the physical location of Zhongshuang 11 in the BnPIR database (http://cbi.hzau.edu.cn/bnapus/index.php) [37] of Huazhong Agricultural University, all genes around the significant SNP (based on the attenuation distance of chromosome LD) of each important locus were determined [34]. All genes in the candidate region were annotated with NR (https://www.biostars.org/p/235632/) and Swiss databases (http://www.swisslifesciences.com/) [38]. The homologous gene of the target gene in *Arabidopsis thaliana* and its function were found on the Tair website (https://www.arabidopsis.org/index.jsp). The tissue-expression levels of the corresponding candidate genes in Zhongshuang 11 were obtained by searching the BnTIR website (http://yanglab.hzau.edu.cn/BnTIR), and the heat map was drawn by Excel software.

### 2.8. Transcriptome Sequencing and Identification of Differentially Expressed Genes

Azuma (hemicellulose content 29.03%) and aphid-resistant rape (hemicellulose content 17.32%) [29], which had a significant difference in hemicellulose content, were selected for stem-tissue sampling. Tissue samples for transcriptome sequencing were taken from 60 cm above the ground thirty days after flowering. Three replicates were taken from each sample, wrapped in foil paper and quickly frozen in liquid nitrogen, and stored at −80 °C

for later use. We extracted total RNA from stalk using the RNAsimple Total RNA Kit (Tiangen Biochemical Technology Co., Ltd., Beijing, China) according to the manufacturer's guidelines. After library preparation, RNA-seq was executed on the Illumina HiSeq 4000 platform. Quality control and normalization of RNA-seq reads were performed using fastqc software and MultiQC, respectively. After the quality was filtered, the RNA-seq reads were compared to the reference genome (ZS11 v20200127) by HISAT2 2.1.0 and then the new transcripts were assembled with the StringTie v1.3.6 software. The gene expression level was quantified with RSEM v1.2.31. The accuracy and sensitivity of sample mutation was detected with SAMTOOLS MPILUP based on the data after the comparison between HISAT2 and genome, and the detection results were stored in VCF format. Then, the R language was used to process the data and draw the heat map. R package (rio and data.table) were used for data importation and data-frame conversion, and R package (stringr) was used for string extraction and standardization and the arrangement of the variance of each row. Then, R package (pheatmap) was used to draw heat mapping of the top 1000 genes with different expression levels. R package DESeq was used for differential-expression analysis, with $p$ value set to 0.05, and R packages (ggplot2 and ggtree) were used for drawing the volcano mapping of differential-expression genes. The extracted differential information was analyzed by GO, and the differential-expression genes were enriched by R package enrich GO, and the GO enrichment map was drawn by R package GOChord.

## 3. Results and Analysis

### 3.1. Evaluation of Hemicellulose Content in Rapeseed Core Natural Population

The hemicellulose content in the stalk and taproot of 139 rapeseed core germplasm resources in three environments was evaluated by near-infrared spectroscopy. In the Zunyi environment, the variation range of hemicellulose in the stalk was 12.56% to 20.75%, and that in taproot was 11.98% to 20.22%. In the Yangluo environment, the variation range of hemicellulose in stalk was 12.02% to 18.68%, and that in taproot was 11.11% to 18.93%. In the Lu'an environment, the variation range of hemicellulose in stalk was 14.29% to 20.27%, and that in taproot was 13.87% to 18.86% (Table 1). The lowest hemicellulose content in the taproot was only 11.11% in the Yangluo environment, while the highest hemicellulose content in the stalk was 20.75% in the Zunyi environment. The hemicellulose content in the stalk and taproot of each environment varied from 5% to 8.24%, with the variation value accounting for 1/3 to 1/2 of the average value, and the variance was from 1.00 to 2.88, all of which were within normal distribution, although the hemicellulose content in each environment was quite different (Table 1, Figure 1). The difference between hemicellulose contents in the stalk and in the taproot of rapeseed in Zunyi was the largest, while that in Lu'an was the smallest, which indicates that environmental factors have a certain influence on hemicellulose content, especially on that in stalk (Table 1). The phenotypic data varied continuously in all environments, and the frequency distribution of the data was roughly a normal distribution, indicating that hemicellulose content is a quantitative trait. Pearson association between hemicellulose content in different environments was calculated (Figure 1). The results showed the hemicellulose content in the stalk and taproot were positively correlated in all of the three environments.

**Table 1.** Statistical analysis of hemicellulose content in stalk and taproot of 139 accessions of rapeseed in different environments. Significance for the effect of genotype (G), environment (E), and genotype by environment interaction (G × E) on phenotypic variance estimated by ANOVA across three environments. (* and ** indicate significant levels of 0.05 and 0.01, respectively).

| Position | Site | Min (%) | Max (%) | Amplitude (%) | Average (%) | Variance | Skewness | Kurtosis | G | E | G × E |
|---|---|---|---|---|---|---|---|---|---|---|---|
| Stalk | Zunyi | 12.56 | 20.75 | 8.19 | 16.61 | 2.88 | 0.00 | −0.09 | | | |
| | Yangluo | 12.02 | 18.68 | 6.66 | 15.53 | 1.87 | −0.18 | −0.32 | 0.000 ** | 0.039 * | 0.758 |
| | Lu'an | 14.29 | 20.27 | 5.98 | 16.67 | 1.00 | 0.59 | 1.22 | | | |
| Taproot | Zunyi | 11.98 | 20.22 | 8.24 | 16.56 | 2.83 | −0.06 | −0.39 | | | |
| | Yangluo | 11.11 | 18.93 | 7.83 | 15.21 | 2.86 | −0.07 | −0.59 | 0.001 ** | 0.002 ** | 0.025 * |
| | Lu'an | 13.86 | 18.86 | 5.00 | 16.13 | 1.18 | 0.28 | −0.29 | | | |

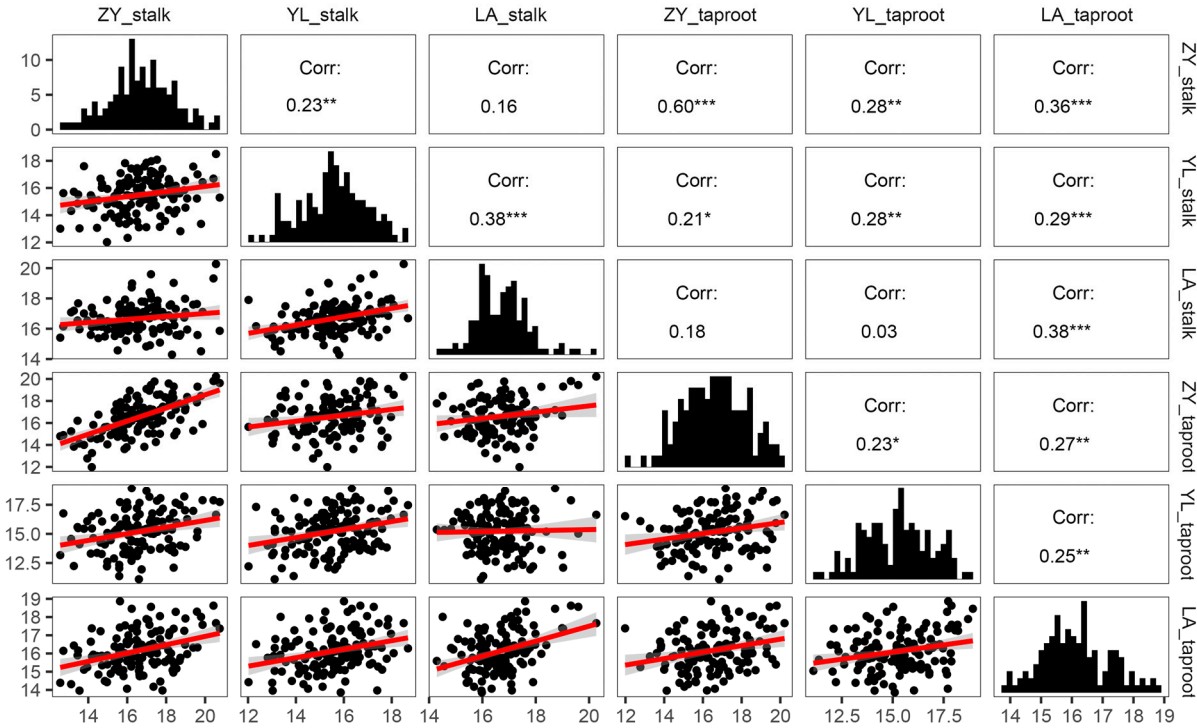

**Figure 1.** Association analysis of hemicellulose content in different tissues of 139 accessions in different environments. The diagonal line is the bar graphs of the hemicellulose content, the lower left corner is the dotted graph of the hemicellulose content and the upper right corner is the association coefficient between the hemicellulose content in each region and part. Corr means correlation coefficient. *, ** and *** indicate significant correlation at the levels of 0.05, 0.01 and 0.001, respectively.

### 3.2. GWAS for Hemicellulose Content

The Manhattan diagram and Q-Q diagram of GWAS results are shown in Figure 2. When the *p* value was 0.001 (−log10 value was 3), a total of 28 SNPs significantly associated with hemicellulose content were identified (Supplementary Table S1); 21 SNPs were detected in the stalk and 7 SNPs were detected in the taproot, with a total of 23 SNPs in the mapping interval, but only 6 SNPs were significantly associated with the two highly expressed genes (Figure 2). SNPs detected in the stalk were mainly concentrated in qHCs.C02, qHCs.C05 and qHCs.C08, and SNPs detected in the taproot were mainly concentrated in qHCt.A09, qHCt.C05 and qHCt.C08. PopLDdecay software was use to draw the LD attenuation diagram to show the degree of chromosome linkage disequilibrium (the longest decay distance was 2500 bp, the break was 6000 bp, and the bin 1 1000-bin 2 6000-break 6000), all genes around the significant SNP of each important locus were determined.

For the stalk correlation interval, the peak value of the qHCs.C02 chromosome was between 56,745,960 bp and 57,991,329 bp, corresponding to Bn-scaff_22144_1-p207843; the peak value of the qHCs.C05 chromosome was between 1,242,999 bp and 9,322,094 bp, corresponding to Bn-scaff_16045_1-p235411; and the peak value of the qHCs.C08 chromosome was between 46,627,013 bp and 46,843,716 bp, corresponding to Bn-scaff_16197_1-p2621761. The contribution rates of phenotypic variance of the three peak markers in the stalk were 17.20%, 10.62% and 8.80%, respectively. For the correlation interval of the taproot, the peak of the qHCt.A09 chromosome was between 6,620,400 bp and 7,024,760 bp, corresponding to Bn-A09-p5255753; the peak of the qHCt.C05 chromosome was between 1,242,999 bp and 1,243,397 bp, corresponding to Bn-scaff_16414_1-p1114696; and the peak of the qHCt.C08 chromosome was between 2,246,8415 bp and 2,246,8464 bp, corresponding to Bn-scaff_16545_1-p80303. The contribution rates of phenotypic variance of the three peak markers in the taproot were 9.49%, 9.18% and 13.10%, respectively. Comparing the correlation interval of hemicellulose content in the stalk and taproot, the overlapping interval between the SNP peak correlation region of qHCs.C05 in the stalk and qHCt.C05 in the taproot was revealed (Table 2), which indicated the high possibility that the hemicellulose content in the stalk and taproot was regulated by common genes.

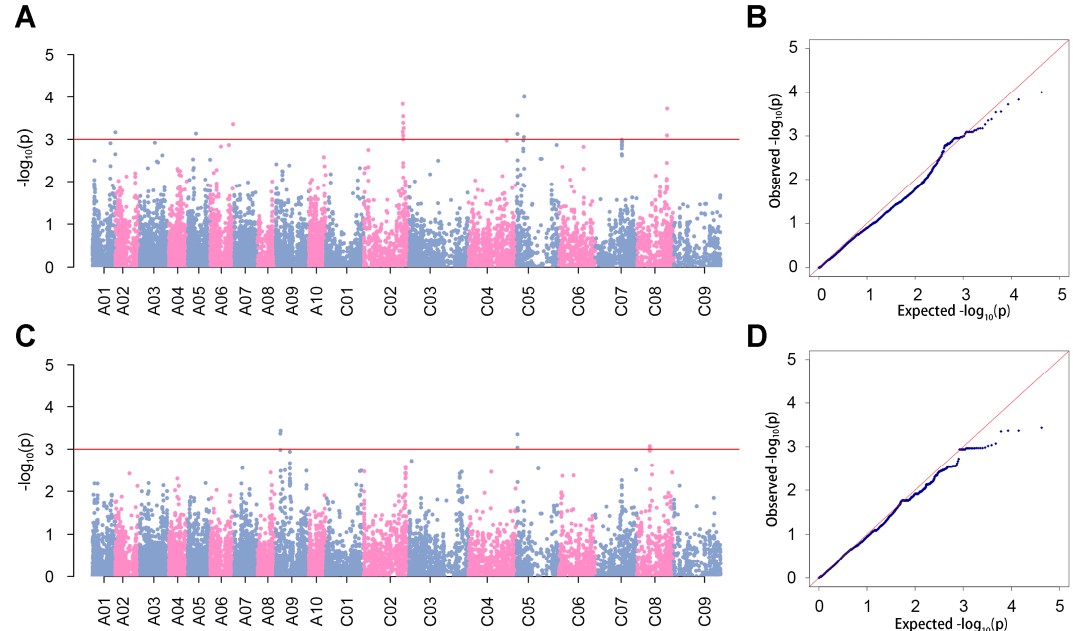

**Figure 2.** GWAS analysis for hemicellulose contents in stalk and taproot of *Brassica napus*. (**A**) Manhattan diagram of stalk hemicellulose, (**B**) QQ diagram of stalk hemicellulose, (**C**) Manhattan diagram of taproot hemicellulose and (**D**) QQ diagram of taproot hemicellulose.

**Table 2.** Information of associated area for hemicellulose content in stalk and taproot of *Brassica napus*.

| Position | Correlation Interval | Originate | Terminate | PeakSNP | −log10(p) | Contribution Rate |
|---|---|---|---|---|---|---|
| Stalk | qHCs.C02 | 56,745,960 | 57,991,329 | Bn-scaff_22144_1-p207843 | 3.84 | 17.20% |
| | qHCs.C05 | 1,242,999 | 9,322,094 | Bn-scaff_16045_1-p235411 | 4.01 | 10.62% |
| | qHCs.C08 | 46,627,013 | 46,843,716 | Bn-scaff_16197_1-p2621761 | 3.72 | 8.80% |
| Taproot | qHCt.A09 | 6,620,400 | 7,024,760 | Bn-A09-p5255753 | 3.44 | 9.49% |
| | qHCt.C05 | 1,242,999 | 1,243,397 | Bn-scaff_16414_1-p1114696 | 3.35 | 9.18% |
| | qHCt.C08 | 22,468,415 | 22,468,464 | Bn-scaff_16545_1-p80303 | 3.07 | 13.10% |

### 3.3. Screening Candidate Genes Associated with Hemicellulose Content

Candidate genes were screened from three loci associated with the stalk and taproot, and the range of candidate regions was determined according to the average LD attenuation distance of chromosome. When contribution rate < 0.2, the LD attenuation of the whole population was 2000 kb. The area with significant LD and its upstream and downstream 2000 kb was determined as the candidate area. According to the annotation information of the database, 11 candidate genes which might be related to hemicellulose synthesis were screened out from the stalk and taproot (Table 3). Subsequently, the gene expression data of the above candidate genes in the Zhongshuang 11 cultivar was obtained on the website (Figure 3), among which the expression levels of *BnaC05G0112100ZS*, *BnaC05G0112200ZS*, *BnaC08G0415400ZS* and *BnaC08G0116500ZS* in the stalk of Zhongshuang 11 were extremely low. Similarly, the expression levels of *BnaC05G0014900ZS* and *BnaC05G0161900ZS* were very high in the stem (Figure 3). RPKM of *BnaC05G0014900ZS* in the stem was over 40 and RPKM of *BnaC05G0161900ZS* was over 248 (obtained the data from the BnTIR website). *BnBXL2.C05* encodes a β-xylosidase protein similar to that in the extracellular matrix, and was a member of the glycosyl hydrolase family 3. *BnGLZ1.C05* encodes a putative glycosyltransferase, which contributes to the biosynthesis of xylan, and its gene expression showed good co-variation with the IRX3 gene which is involved in secondary cell wall synthesis. The remaining five candidate genes (*BnaC02G0468200ZS*, *BnaC05G0092200ZS*, *BnaC05G0112300ZS*, *BnaC05G0112400ZS* and *BnaC05G0156900ZS*) were xyloglucan oxo-acetyltransferase, xyloglucan endoglucanase, glycosyltransferase and galactosyltransferase, which may regulate the biosynthesis of hemicellulose.

**Table 3.** Information of candidate genes associated with hemicellulose content in stalk and taproot of *Brassica napus*.

| Candidate Gene | Originate | Terminate | Arabidopsis Homologous Gene | Gene Name | Functional Annotation |
|---|---|---|---|---|---|
| *BnaC02G0468200ZS* | 57,411,375 | 57,412,944 | *AT3G28150* | *AXY4L* | Involved in the synthesis and deposition of secondary wall cellulose |
| *BnaC05G0014900ZS* | 1,016,442 | 1,019,787 | *AT1G02640* | *BXL2* | Encodes a protein similar to a beta-xylosidase |
| *BnaC05G0092200ZS* | 5,255,987 | 5,257,234 | *AT1G11545* | *XTH8* | Xyloglucan endotransglucosylase/hydrolase 8 |
| *BnaC05G0112100ZS* | 6,929,662 | 6,931,370 | *AT1G14080* | *FUT6* | Encodes an alpha-(1,2)-fucosyltransferase |
| *BnaC05G0112200ZS* | 6,932,039 | 6,934,614 | *AT1G14080* | *FUT6* | Encodes an alpha-(1,2)-fucosyltransferase. |
| *BnaC05G0112300ZS* | 6,937,873 | 6,939,470 | *AT1G14100* | *FUT8* | Member of glycosyltransferase Family- 37. |
| *BnaC05G0112400ZS* | 6,944,599 | 6,946,246 | *AT1G14100* | *FUT8* | Member of glycosyltransferase Family- 37 |
| *BnaC05G0156900ZS* | 10,051,053 | 10,052,411 | *AT1G18690* | *GMA12* | Galactosyl transferase GMA12/MNN10 family protein |
| *BnaC05G0161900ZS* | 10,585,900 | 10,586,958 | *AT1G19300* | *GLZ* | Encodes a putative family 8 glycosyl transferase that contributes to xylan biosynthesis |
| *BnaC08G0415400ZS* | 46,598,950 | 46,599,504 | *AT3G62720* | *XT1, XXT1* | Encodes a protein with xylosyltransferase activity |
| *BnaC08G0116500ZS* | 21,355,187 | 21,356,233 | *AT4G14130* | *XTH15, XTR7* | Xyloglucan endotransglycosylase-related protein |

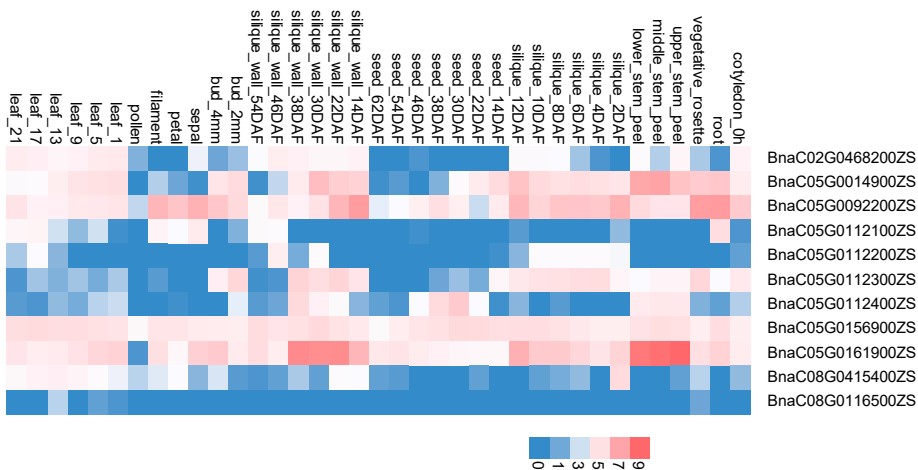

**Figure 3.** Expression heat map of candidate genes for hemicellulose content in stalk and taproot of *Brassica napus*. The expression amount is transformed by $\log_{10}(x + 1)$.

### 3.4. Identification of Differentially Expressed Genes and GO Enrichment Analysis

At the stage of 30 days after flowering, stalk samples of two very different hemicellulose content accessions Azuma (rapeseed accession with high hemicellulose, 29.03%) and aphid-resistant rape (rapeseed accession with low hemicellulose, 17.32%) were sent for transcriptome sequencing analysis. The results showed that there was a significantly high number differentially expressed genes detected between the two materials. In the stems of the two extreme materials, 22,159 significantly different genes, including 10,439 upregulated and 11,720 down-regulated genes, were detected (Figure 4A). These mainly belonged to the glycoside hydrolase and TBL gene families. Through the GO function enrichment analysis of the first 1000 differentially expressed genes, it was found that the enrichment genes were mainly in plastids, cytosol and chloroplasts, and the differential transcription factors were mainly involved in biological processes of translation and signal transduction and regulating molecular functions of DNA binding and RNA binding (Figure 4B). According to the functional annotation of *Arabidopsis* homologous genes, 27 differentially expressed genes were screened out (Supplementary Table S2), and the function of these genes was basically related to cell-wall formation. The expression difference of *BnaC09G0019300ZS* (*AT4G03210*) between the two extreme materials was found to be the most significant of the 26 genes. It was also revealed that this gene functioned in cell-wall loosening and rearrangement and only acted on the shoot-tip area through acquiring the function of the gene in *Arabidopsis thaliana*. The other three genes, *BnaC02G0539800ZS*, *BnaC03G0005500ZS* (*AT5G64020*) and *BnaC02G0002600ZS* (*AT5G01360*), have been proved to be involved in the cellulose synthesis and deposition in secondary wall [39]. In addition, *BnaA07G0164200ZS* (*AT2G28760*) and *BnaA07G0142000ZS* (*AT5G67230*) are involved in hemicellulose biosynthesis. Therefore, the differentially expressed genes in rapeseed stems are mainly involved in cell-wall development, which might improve the toughness of stems.

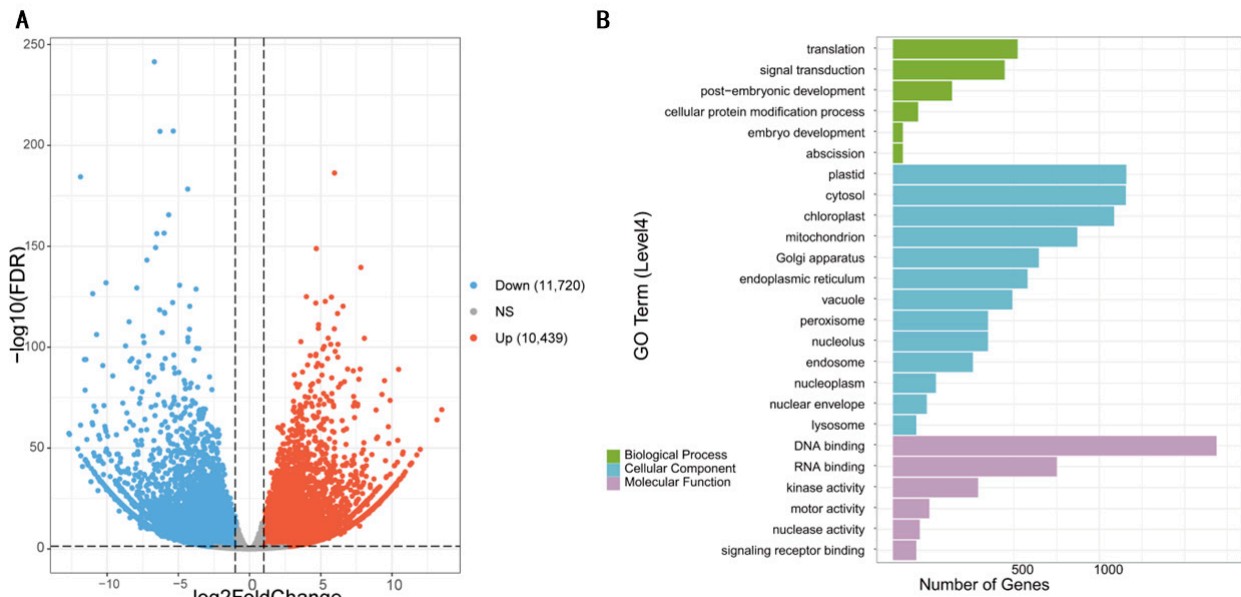

**Figure 4.** Identification and GO enrichment analysis of differentially expressed genes among materials with extreme hemicellulose content in rapeseed stalk. (**A**) Volcano map of all gene-expression differences among materials with extreme hemicellulose content in rapeseed stalk. The blue dot on the left is down-regulated genes, and the red dot on the right is up-regulated genes. (**B**) GO enrichment of differentially expressed genes.

### 3.5. Combining GWAS and Transcriptome to Identify Candidate Genes

The 27 differentially expressed genes obtained from transcriptome data analysis, were compared with the candidate genes located by GWAS, but no common candidate genes were found. Then, the differentially expressed genes existing in the associated region were further compared with the significantly high number differentially expressed genes in transcriptome data (Supplementary Table S3), and eight candidate genes with significant differential expression were screened out. According to the functional annotation of *Arabidopsis* homologous genes (Table 4), two key candidate genes associated with hemicellulose content, *BnaC05G0092200ZS* and *BnaC05G012400ZS* (*BnXTH8.C05* and *BnFUT8.C05*), were finally obtained.

**Table 4.** Differentially expressed genes in the associated region of hemicellulose content in rapeseed stalk.

| Candidate Genes | Homologs in Arabidopsis | Gene Annotation | Log2FC |
|---|---|---|---|
| *BnaC02G0468300ZS* | *AT3G28180* | Encodes a gene similar to cellulose synthase | −4.71 |
| *BnaC02G0470400ZS* | *AT3G28740* | Encodes a member of the cytochrome p450 family | 7.48 |
| *BnaC05G0070100ZS* | *AT1G09310* | ABA responsive trichome formation regulator | −7.73 |
| *BnaC05G0089100ZS* | *AT1G11080* | Serine carboxypeptidase-like 31 | 7.77 |
| *BnaC05G0092200ZS* | *AT1G11545* | Xyloglucan endotransglucosylase/hydrolase 8 | −2.99 |
| *BnaC05G0112400ZS* | *AT1G14100* | Member of Glycosyltransferase Family- 37 | −3.68 |
| *BnaC05G0121900ZS* | *AT3G30739* | Hypothetical protein | 7.30 |
| *BnaC05G0136700ZS* | *AT1G16410* | Member of CYP79F, Encodes cytochrome P450 | −10.74 |

## 4. Discussion

Under the double pressures of scarcity of fossil resources and deterioration of the ecological environment, biomass has the double advantages of zero emission of carbon dioxide and regeneration, so the production of biofuels and chemicals from biomass has become a hot topic in all countries [40]. Compared with cellulose, hemicellulose is highly branched and amorphous, and its stability and polymerization degree are lower.

Moreover, the content of hemicellulose in biomass is second only to that of cellulose [15,41], so hemicellulose is more easily used in industry than cellulose and lignin. Rapeseed stalk is rich in hemicellulose, after processing, can be used as biomass energy instead of fossil energy, helping to alleviate the shortage of non-renewable resources and providing favorable conditions for the high-value utilization of rapeseed stalk as biomass energy.

Hemicellulose is the second major component of plant cell wall, and is a regulator of cell-wall formation [24]. By measuring the hemicellulose content of rapeseed in different environments, it was found that the hemicellulose content in stalk and taproot of rapeseed are positively correlated. Meanwhile, through GWAS analysis, a common locus was located in C05. This locus links the hemicellulose content in the stalk and taproot, indicating that this locus may regulate the hemicellulose content in the stalk and taproot at the same time (Table 2, Figure 2). The high hemicellulose content in the taproot may help rapeseed to resist root lodging, while the high hemicellulose content in the stalk may help rapeseed to resist stem bending [24]. It is therefore possible to increase hemicellulose content to improve cell-wall toughness by mining key common genes, which may not only increase the bending resistance of stalks, but also greatly increase the root-lodging resistance. Chemical components in different parts of maize were determined, and it was found that the hemicellulose content and cellulose content in straw had the same changing trend, while lignin content was negatively correlated with cellulose content [42]. Therefore, hemicellulose content was negatively correlated with lignin content, and lignin was the key substance to support the weight of crops above ground. By marker-assisted selection, researchers have developed a rice cultivar named Yexing with the elite traits of low lignin, long stalks, strong lodging resistance and high biomass [43]. It was concluded that higher hemicellulose content may reduce lignin content and increase lodging resistance, but this is inconsistent with previous studies which showed that lignin can increase stalk-bending resistance [44]. Therefore, it is necessary to further study whether the candidate gene can improve the hemicellulose content of crops, enhance the resistance to root lodging and bending, and improve the yield of rapeseed.

Xylan is one of the richest polysaccharides in hemicellulose in the primary cell wall of higher plant [45]. It forms connections between microfibers, accounting for 50% of the total amount in plant woody tissues, and constitutes hard cells [46]. By comparing the correlation intervals of hemicellulose content in the stalk and taproot of rapeseed in different environments with genome-wide association studies, it was found that *BnaC02G0468200ZS* was significantly associated with hemicellulose content in the stalk, and played an important role in cell-wall structure and normal plant development [47]. Interesting, it is very likely that the same gene on chromosome C05 is located in the taproot and stalk. *BnBXL2.C05* and *BnGLZ1.C05* are located in the associated region, and their expression levels are high in specific tissues. The former can alleviate the inhibition of xylan on xylanase and cellulase [48]. The latter can promote the biosynthesis of xylan and participate in the synthesis of pectin at the same time. It has been observed that the deletion of this gene will not affect the content of hemicellulose in the body, but it can change the binding force of the plant cell wall to aluminum, thus affecting the sensitivity to aluminum [49]. The two related and highly expressed genes can promote the accumulation of xylan, and alleviate the inhibition of related enzymes, thus improving its economic value. At present, there is little research on glycosyltransferase, other that the finding of about ten gene families in animals and plants. This enzyme participates in the formation of plant cell walls and plays an important role in the growth and development of plants. Different glycosyl modification patterns lead to different xylans synthesized in different plants. Therefore, xylan, as one of hemicellulose polysaccharides, may play a very important role in rapeseed.

It is difficult to accurately determine effective sites through GWAS. In particular, the ability to detect sites controlled by micro-effect polygenes is insufficient. Therefore, combining GWAS and RNA-seq analysis can improve the resolution efficiency to mine the key genes or regulatory networks for complex agronomic traits of plants [50]. In the present study, 27 genes which are associated with hemicellulose content and with significant differ-

ential expression in two extreme materials, were screened through transcriptome analysis (Supplementary Table S2). By querying the functions of homologous genes in *Arabidopsis thaliana*, it was found that the functions of most of the genes were related to cell walls. Among them, *BnaC09G0019300ZS*, the gene with the highest differential expression, acted on the stems and pedicels at the tips, and played a role in the loosening and rearrangement of cell walls. At the same time, three genes were found on chromosome qHCs.C02, which had been proved to be related to cellulose synthesis and deposition in the secondary cell wall. There are two other genes involved in the biosynthesis of xylan and xyloglucan, both located on chromosome qHCs.A07. By comparing the candidate genes screened by GWAS with the differentially expressed genes screened by transcriptome data analysis, it was found that the candidate genes obtained by the two methods did not coincide. However, by increasing the screening differential value, we searched for the differentially expressed genes in the candidate interval located by GWAS, and found that some genes were related to membrane proteins, which might regulate the hemicellulose content and function of the cell wall. At the same time, it was found that *BnaC05G0092200ZS* (*BnXTH8.CO5*) and *BnaC05G0112400ZS* (*BnFUT8.CO5*) genes were in the candidate interval located by GWAS. According to the functional annotation of *Arabidopsis thaliana* homologous genes, it is known that *BnaC05G0092200ZS* (*AT1G11545*) belongs to *XTH8* gene family, which codes for xyloglucan transglycosidase/hydrolase 8. It is a type of cell wall relaxation enzyme, which regulates cell growth and plant development by regulating SA accumulation [51], and it plays an important role in cell-wall remodeling, affecting plant growth and development. *BnaC05G0112400ZS* (*AT1G14100*) is a member of the gene family of glycosyltransferases (*FUT8*). Glycosyltransferases are connected to different receptor molecules in organisms through catalytically activated sugars, and the connected receptor molecules play different functions. Therefore, glycosylated products have many biological functions. All the identified genes are involved in the important regulation pathway of hemicellulose biosynthesis, which also indicates that hemicellulose content be a quantitative trait controlled by multiple genes.

## 5. Conclusions

The hemicellulose in rapeseed plays an important role in organ development. In this study, six QTL of hemicellulose content in the rapeseed stalk and taproot across three different environments were screened by genome-wide association study. Meanwhile, two key candidate genes underlying QTL, *BnaC05G0092200ZS* and *BnaC05G0112400ZS*, which regulate hemicellulose synthesis, were discovered by RNA-seq. These QTL and candidate genes provided the genetic basis and target variations of selecting rapeseed lines with high hemicellulose content. In a follow-up experiment, by analyzing the functions of two key candidate genes and measuring the hemicellulose content, we can study the tolerance of rapeseed with high hemicellulose content in roots to metal ions, the relationship between high hemicellulose content in stalk and the bending resistance of stems, and decipher the influence of high hemicellulose content in rapeseed roots and stalks on crop production and energy utilization. This will have significance for bioenergy production and high crop yield. Further deciphering the biological and genetic basis of hemicellulose in rapeseed will be of great value in elite variety development by marker-assisted selection, in dissection of the cell-wall formation mechanism and in highlighting bioenergy utilization.

**Supplementary Materials:** The following supporting information can be downloaded at: https://www.mdpi.com/article/10.3390/agronomy12112886/s1. Supplementary Figure S1: Temperature changes of rapeseed planted in three environments; Supplementary Table S1: Genome-wide association study results of hemicellulose content in different tissues of 139 rapeseed samples in different environments; Supplementary Table S2: Significant differentially expressed genes associated with hemicellulose in rapeseed stalk and taproot; Supplementary Table S3: Differentially expressed genes in rape stems in related regions; Supplementary Table S4: The 139 rapeseed core germplasm resources with hemicellulose content measured.

**Author Contributions:** J.L. and W.W. designed and supervised the study; Y.X. and J.L. collected the samples; L.L. provided the technique guide for hemicellulose analysis; Q.H. provided the germplasms; Y.Y. and W.Y. performed transcriptome analyses and all data analysis; Y.X., W.W., and J.L. wrote the manuscript and revised the manuscript. All authors have read and agreed to the published version of the manuscript.

**Funding:** This research was funded by the Natural Science Foundation of China (U19A2029), the Central Public-Interest Scientific Institution Basal Research Fund (No. 161017202203), the Key Research Program & Technology Innovation Program of the Chinese Academy of Agricultural Sciences (CAAS-ZDRW202105) and Technological Innovation and Development Projects of Hubei (2021BLB159). And The APC was funded by the Natural Science Foundation of China (U19A2029)

**Institutional Review Board Statement:** Not applicable.

**Informed Consent Statement:** Not applicable.

**Data Availability Statement:** Sequence data from this article can be found in the Genome Sequence Archive (GSA) repository of the National Genomics Data Center (NGDC), Beijing, China (Database: https://ngdc.cncb.ac.cn/gsub/) (24 August 2022), under the accession number PRJCA011375.

**Conflicts of Interest:** The authors declare no conflict of interest.

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
