# Peer review of "Dissecting the Genetic Mechanisms of Hemicellulose Content in Rapeseed Stalk"

_agronomy, doi:10.3390/agronomy12112886_

Round 1
Reviewer 1 Report
Dear Authors and editor:
It is my pleasure that I reviewed this manuscript " Dissecting the genetic mechanisms of hemicellulose content in rapeseed stalk”. The manuscript is great and appealing. However, there are some major and minor issues pending, which need to be addressed in the revision.
11 The abstract need to include the number of genotypes used, percentage of phenotypic variation explained by the significant SNPs. The two key genes identified by both GWAS and TWAS, what percentage of phenotypic variation expressed by the significant SNPs from GWAS.
22 Line 47: Add reference
33 Line 51-56: Add reference
44 Line 60: “researchershave” typo. researchers have. More so, reference to these studies won’t be a bad idea
55 Line 85 to 89: Add reference
66 The introduction needs to be tightened up a little. There are couples of loos transition.
77 Line 234 to 242: Please check for consistency. “13.87%~18.86%”. Here no space between the tilde. In some instance tilde was used while in another instance “to” was used: 5% to 8.24%
88 Line 244: 1.003 to 2.879. For consistency I will encourage to use specific decimal place throughout the text.
99 Line 245 to 250: Cite table or Figure for the results you illustrating here.
110. Line 332: What statistical test conducted to compare results from TWAS and GWAS? Fisher combined test is one of such that can be used
111. Line 342 to 345: Add reference
112. Line 360 to 367: Add reference
113. Line 404 to 405: Add reference
114. The discussion needs to be well synchronized with the results. More effort is required in the discussion section and needs to link findings with previous studies.
115. The discussion should be separated from conclusion
116. In general, kindly add more reference.
Author Response
Response to the Review 1#
11 The abstract need to include the number of genotypes used, percentage of phenotypic variation explained by the significant SNPs. The two key genes identified by both GWAS and TWAS, what percentage of phenotypic variation expressed by the significant SNPs from GWAS.
Reply: Thanks for the good suggestion. We have added the number of genotypes, significant percentage of phenotypic variation explained by SNPs and identified two key candidate gene IDs in the abstract as suggestion.
22 Line 47: Add reference
33 Line 51-56: Add reference
44 Line 60: “researchershave” typo. researchers have. More so, reference to these studies won’t be a bad idea
55 Line 85 to 89: Add reference
Reply: Thanks. We have added the references in the corresponding sentences, and added them to the Reference part in order.
66 The introduction needs to be tightened up a little. There are couples of loos transition.
Reply: Thanks for the professional suggestion. We revised the introduction part and deleted the redundant sentences.
77 Line 234 to 242: Please check for consistency. “13.87%~18.86%”. Here no space between the tilde. In some instance tilde was used while in another instance “to” was used: 5% to 8.24%
Reply: Thanks for proposed the format error. We have made a unified revision of changing "~" into "to" in the thorough draft.
88 Line 244: 1.003 to 2.879. For consistency I will encourage to use specific decimal place throughout the text.
Reply: Thanks. We unify the format in the revised draft into two significant digits after the decimal point.
99 Line 245 to 250: Cite table or Figure for the results you illustrating here.
Reply: Thanks. We have added the table and figure for the result in the revised draft.
- Line 332: What statistical test conducted to compare results from TWAS and GWAS? Fisher combined test is one of such that can be used.
Reply: Thanks for the advice. We only compared the QTL positions and underlying genes by GWAS with the differential expression genes by transcriptome, and excavated the common candidate genes. We did not compare the results by Fisher combined test.
- Line 342 to 345: Add reference
- Line 360 to 367: Add reference
- Line 404 to 405: Add reference
Reply: Thanks. We have added the references in the corresponding sentences, and added them to the Reference part in order.
- The discussion needs to be well synchronized with the results. More effort is required in the discussion section and needs to link findings with previous studies.
Reply: Thanks to the professional advice. We have revised the discussion part and concise and clarify the ambiguous sentences and paragraph synchronized with the results part.
- The discussion should be separated from conclusion
Reply: Thanks. We reorganized these parts as the suggestion and separated the two parts.
- In general, kindly add more reference.
Reply: Many thanks. We have added the more than 5 references according to the reviewer`s suggestion in the revised manuscript.

Author Response
Response to the Review 2#
Line 35: “Further RNA-seq analysis showed that two key differentially expressed genes involved in hemicellulose synthesis were identified, which were consistent with the candidate genes
located by GWAS“. Sentence not clear since we do not know between what and what the DEGS
are identified.
Reply: Thanks for pointing out the problem. We have reorganized the sentence and delete “which were consistent with the candidate genes located by GWAS” and make it clear.
Line 37: “This study excavated the key sites and candidate genes for regulating hemicellulose synthesis, which provided a theoretical basis for developing rapeseed varieties with high hemicellulose content, which improves the rapeseed cultivars with high-quality lodging- resistant and also highlight the value as resources of bioenergy industry”. Too long of a sentence, it’s hard to follow. Please rephrase or cut the sentence in half.
Reply: We have rephrased this sentence. Changing into “This study excavated the key loci and candidate genes for regulating hemicellulose synthesis, which provided a theoretical basis for developing rapeseed varieties with high hemicellulose content. At the same time, our results will be helpful to improves the rapeseed cultivars with high lodging-resistant and also highlight the value as resources of bioenergy industry.”.
Line 60: Space is missing between “researchers” and “have”
Reply: We have inserted spaces between “researchers” and “have”.
Line 66: “Hemicellulose is synthesized in Golgi apparatus, deposited on the surface of cell wall
through vesicles, and can be dissolved in alkaline solution” The reference is missing.
Reply: Thanks. We have added the references in the corresponding sentences, and added them to the Reference part in order.
Line 68: “Do not understand why the authors put this sentence there: “Hemicellulose is high value as resources of chemical industry and bioenergy industry. “It might be preferable to put in the next paragraph?
Reply: Thanks for the good suggestion. We have moved it in the next paragraph.
Line 70: Not really clear what the authors meant when they wrote:” The hemicellulose content
in plants is second only to cellulose”. Please clarify.
Reply: Thanks. This sentence is meaningless here. So this sentence has been deleted in order to make the structure more compact and reasonable.
Line 88: Space is missing.
Reply: We have inserted spaces here.
Line 100: “The XTH31 gene have been found which can regulate the content of xyloglucan in cell wall hemicellulose and the aluminum sensitivity of Arabidopsis thaliana (Wan et al., 2018; Zheng, 2014). “The beginning of the sentence is weird. “The XTH31 gene have been found to regulate…” Might be more appropriated.
Reply: Thanks. We have revised this sentence.
Line 106: “Therefore, by increasing the hemicellulose content in crop roots, improving the capacity of crops to adsorb heavy metals in soil, reducing the content of heavy metals in soil and inhibiting the transport of heavy metals to the shoots of plants, it is helpful to improve crop yield and quality.” Really long and repetitive sentence. Rephrase please for more clarity and being more straight forward.
Reply: Thanks. We have rephrased this sentence. Changing into “Thereby reducing the content of heavy metals in soil, effectively reducing heavy metal stress and improving crop yield. Therefore, by increasing the hemicellulose content in crop roots, reducing the content of heavy metals in soil, it is helpful to improve crop yield and quality.”.
Line 143: “The genetic basis of hemicellulose content in rapeseed straw has not been reported
so far.” And how about in Stalk?
Reply: Thanks. What we want to say is that there are seldom reports on stalk, and the wrong expression has been corrected.
Line 149: “Two key candidate genes associated with hemicelllose synthesis in rapeseed stem were obtained by analyzing transcriptome data of rapeseed stem.” Still unclear comparing what and what these genes have been identified.
Reply: Thanks. We have added gene ID, and here we only briefly summarize the conclusions obtained by the methods used. The sentence rephrased: “Two key candidate genes BnaC05G0092200ZS and BnaC05G0112400ZS associated with hemicellulose synthesis were obtained by analyzing transcriptome data of rapeseed stem.”
Line 294: avoid repetition. and “the expression levels of BnaC05G0014900ZS (BnBXL2.C05) and BnaC05G0161900ZS (BnGLZ1.C05) were very high” Not supported by the data. Please provide data to support this assumption.
Reply: Thanks. We have supplemented the relevant data and revised the sentences. “the expression levels of BnaC05G0014900ZS and BnaC05G0161900ZS were very high in stem organs (Figure 3). RPKM of BnaC05G0014900ZS in stem is over 40 and RPKM of BnaC05G0161900ZS is over 248 obtained the data from the BnTIR website.
Line 306: Why these two accessions have been picked up? please provide the rational.
Reply: Thanks. We give the reason in the revised MS. At stage of 30 days after flowering, stalk samples of two extremely differential hemicellulose content accessions Azuma (rapeseed accession with high hemicellulose, 29.03%) and Aphid Resistant Rape (rapeseed accession with low hemicellulose, 17.32%) were taken and sent to the company for transcriptome sequencing analysis.
Line 309: “enormous” is not scientific. “A significantly high number” might be more
appropriate;
Reply: Thanks. We have corrected the inappropriate words and checked the full text.
Line 319: “Should be Table 3 not 2.
Reply: Thanks. We have revised the inappropriate chart insertion and checked the full text.
Line 335: Change “enormous”.
Reply: Thanks. We have corrected the inappropriate words and checked the full text.
